# Three months use of Hybrid Closed Loop Systems improves glycated hemoglobin levels in adolescents and children with type 1 diabetes: A meta-analysis

**Yuan-yuan WANG**[1], **Hui-min YING**[1], **Fang TIAN**[1], **Xiao-lu QIAN**[1], **Zhen-feng Zhou**[2]*

**1** Department of Endocrinology, Xixi Hospital of Hangzhou (Affiliated Hangzhou Xixi Hospital of Zhejiang Chinese Medical University), Hangzhou, Zhejiang Province, Hangzhou, China, **2** Department of Anesthesiology, Hangzhou Women's Hospital (Hangzhou Maternity and Child Health Care Hospital, Hangzhou First People's Hospital Qianjiang New City Campus, Zhejiang Chinese Medical University), Hangzhou, China

☯ These authors contributed equally to this work.

* zhenfeng9853@163.com

## Abstract

### Background

Longer outpatient studies have demonstrated that hybrid closed loop (HCL) use has led to a concomitant reduction in glycated hemoglobin(HbA1c) by 0.3%–0.7%. However, reports have also indicated that HbA1c levels are not declined in the long-term use of HCL. Therefore, we wonder that 3 months use of HCL could improve glycated hemoglobin levels in adolescents and children with T1D.

### Methods

Relevant studies were searched electronically in the Cochrane Library, PubMed, and Embase utilizing the key words "Pediatrics or Child or Adolescent", "Insulin Infusion Systems" and "Diabetes Mellitus" from inception to 17[th] March 2024 to evaluate the performance of HCL on HbA1c in adolescents, and children with T1D.

### Results

Nine studies involving 927 patients were identified. Three months use of HCL show a beneficial effect on HbA1c management ($p < 0.001$) as compared to standard of care in adolescents and children with T1D, without evidence of heterogeneity between articles ($I^2 = 40\%$, $p = 0.10$). HCL did significantly increase the overall average percentage of hypoglycemic time between 70 and 180 mg/dL (TIR) ($p < 0.001$; $I^2 = 51\%$). HCL did not show a beneficial effect on hypoglycemic time <70 mg/dL and <54 mg/dL ($p > 0.05$). The overall percentage of hyperglycemic time was significantly decreased in HCL group compared to the control group when it was defined as >180 mg/dL ($p < 0.001$; $I^2 = 83\%$), >250 mg/dL ($p = 0.007$, $I^2 = 86\%$) and >300 mg/dL ($p = 0.005$; $I^2 = 76\%$). The mean glucose level was significantly decreased by HCL ($p < 0.001$; $I^2 = 58\%$), however, no significant difference was found in

**Data Availability Statement:** All data analysed during this study are collected from published articles [Table 1].

**Funding:** The author(s) received no specific funding for this work.

**Competing interests:** The authors have declared that no competing interests exist.

**Abbreviations:** BMI, body mass index; CI, confidence interval; DKA, diabetic ketoacidosis; HCL, Closed-Loop Systems; MD, mean difference; OR, odds ratio; PLGS, predictive low glucose suspend systems; RCTs, randomized controlled studies; RR, relative risk; SAP, sensor-augmented pump; SMD, standardised mean difference; T1D, type 1 diabetes; TIR, the percent time in range.

coefficient of variation of sensor glucose ($p = 0.82$; $I^2 = 71\%$) and daily insulin dose ($p = 0.94$; $I^2 < 0.001$) between the HCL group and the control group.

## Conclusions

HCL had a beneficial effect on HbA1c management and TIR without increased hypoglycemic time as compared to standard of care in adolescents and children with T1D when therapy duration of HCL was not less than three months.

## Trial number and registry URL

CRD42022367493; https://www.crd.york.ac.uk/PROSPERO, Principal investigator: Zhenfeng Zhou, Date of registration: October 30, 2022.

## Introduction

The overall annual incidence of type 1 diabetes (T1D) was reported to be increasing by around 3% in adolescents and children [1]. Cognition and brain structure can be profoundly impacted if glycemic control is poor controlled in the early course of T1D, particularly in very young children [2, 3]. The 2019 American Diabetes Association guidelines suggest that HbA1c should be control at <7.5% for all children [4]. However, the achievement of this glycemic control goal is affected by several behavioral and developmental factors in adolescents and children [5].

Several new devices have been developed to improve T1D management in recent decades, including real-time continuous glucose monitoring (CGM) devices and other insulin pumps [6], however, the management of HbA1c has actually worsened in adolescents and children over the last 10 years [6–9]. According to the latest ISPAD consensus clinical practice guidelines [10], HCL is strongly recommended for youth with diabetes as it could improve time in range (TIR) by minimizing hypoglycemia and hyperglycemia, and is especially beneficial in the overnight period (level A).

Longer outpatient studies have also demonstrated that HCL use has led to a concomitant reduction in HbA1c by 0.3%–0.7% [11–19]. However, most of the above studies were not randomized clinical trials or HCL administration period was less than 3 months. Previous studies have shown that HbA1c levels decrease in adolescents and adults after at least 3 months of HCL use [20–22]. Furthermore, reports have also indicated that HbA1c levels are not declined in the long-term use of HCL as compared to the control group [23–25].

Recent meta-analysis [26, 27] had also concerned HbA1c of HCL in patients with T1D, however, the recent important studies [8, 9, 11, 22, 24, 28, 29] were not included in those recent meta-analysis [26, 27], we also noticed that one recent meta-analysis [26, 27] included adults with a study period of less than 8 weeks, and the study period of another recent meta-analysis [27] was different from 48 h to 6 months and 40% of follow up were no more than 7days. Therefore, we wonder that at least 3 months use of HCL could improve glycated hemoglobin levels in adolescents and children with T1D.

## Materials and methods

We performed this meta-analysis in accordance with the PRISMA [30] and we registered the protocol with PROSPERO (CRD42022367493).

## Search strategy

The Cochrane Library, PubMed, and Embase databases were searched from inception to 17th March 2024 by two investigators (Fang TIAN and Hui-min YING) to identity relevant studies.

The following key words were used: "Pediatrics or Child or Adolescent", "Insulin Infusion Systems" and "Diabetes Mellitus". A detailed list of search strategies is provided in S1 File. First, two independent reviewers (Fang TIAN and Hui-min YING) excluded irrelevant studies by screening the titles and abstracts. Then, the full texts of the remaining studies were reviewed to ensure that all relevant studies had been evaluated. An independent third author (Xiao-lu QIAN) was consulted to resolve any disagreements between the reviewers.

**Study selection and data extraction.** Only randomized controlled studies (RCTs) in the English language were included. Other eligibility criteria were as follows: (a) adolescents and children with T1D; (b) treatment with closed-loop systems; (c) treatment with insulin pump without closed-loop systems in the control group; and (d) duration of study was not less than 12 weeks. Studies with an adult population (above 21 years old), non-RCTs, conference abstracts, case reports and studies with no control group (standard of care) were excluded.

The HCL system was defined as having glucose-sensing technology and algorithm-based insulin delivery including single-hormone (nonadjustable insulin pump systems) or dual-hormone (insulin and glucagon pump systems). The standard of care was confirmed as any kind of standard care provided including sensor-augmented pump (SAP), open-loop and PLGS (predictive low glucose suspend systems).

Two authors (Fang TIAN and Hui-min YING) identified all studies by the above eligibility and exclusion criteria. We extracted the following data: the type of trial, publication year, sex, age, BMI, duration of diabetes, duration of HCL treatment, sample size and glycemic outcomes (Table 1).

**Outcome definition [31].** The primary endpoint was the HbA1c level in adolescents and children with T1D at the end of the study period. The secondary outcomes were as follows: the percent time in range (TIR) between 70 and 180 mg/dL, percentage of hypoglycemic time (the sensor glucose defined as <70 mg/dL and < 54 mg/dL) and hyperglycemic time (>180 mg/dL, > 250 mg/dL and > 300 mg/dL), sensor glucose values, coefficient of variation of sensor glucose values, and total daily insulin dose requirement. Furthermore, data on adverse events were collected, including (1) severe hypoglycemia: hypoglycemia needing to be treated as altered consciousness; (2) diabetic ketoacidosis (DKA, defined according to the criteria of the Diabetes Control and Complications Trial); and (3) hyperketonemia. Outcomes were also

**Table 1. Main characteristics of the included trials.**

| First author | Year | Journal | Type of Trial | Sample(n) HCL/control | Female(n) HCL/control | Age(mean ± SD, y) HCL | Age(mean ± SD, y) control | BMI(mean ± SD) HCL | BMI(mean ± SD) control |
|---|---|---|---|---|---|---|---|---|---|
| H. Thabit | 2015 | N Engl J Med | randomised crossover | 49(25/24) | 11/11 | 12.0±3.4 | 12.0±3.4 | 18.9±3.5 | 18.9±3.5 |
| Marc D. Breton | 2020 | N Engl J Med | randomized- control | 100(78/22) | 38/12 | 11.3±2.0 | 10.8±2.4 | 0.4±1.0* | 0.5±1.0* |
| Elvira Isganaitis | 2021 | Diabetes Technol Ther | randomized- control | 63(40/23) | 17/10 | 17.0±3.0 | 17.0±3.0 | 29.0±8.9 | 26.0±1.5 |
| Mary B.Abraham | 2021 | JAMA Pediatrics | randomized- control | 135(67/68) | 37/39 | 15.2±3.3 | 15.4±3.0 | 0.7±0.8* | 0.7±0.7* |
| Lauren G.Kanapka | 2021 | Diabetes care | randomized- control | 100(78/22) | 38/11 | 11.6±2.0 | 11.0±2.4 | NA | NA |
| J. Ware | 2022 | N Engl J Med | randomised crossover | 147(73/74) | 31/31 | 5.6±1.6 | 5.6±1.6 | NA | NA |
| Julia Ware | 2022 | The Lancet. Digital health | randomized- control | 133(65/68) | 37/39 | 13.1±2.6 | 12.8±2.9 | 0.35±0.86* | 0.58±0.89* |
| Boughton | 2022 | N Engl J Med | randomized- control | 97(51/46) | 43/25 | 12.0±2.0 | 12.0±2.0 | NA | NA |
| R. Paul Wadwa | 2023 | N Engl J Med | randomized- control | 102(68/34) | 34/19 | 3.84±1.23 | 4.06±1.5 | 0.73±1.28* | 0.69±0.86* |

*:BMI Z-Score

evaluated in based on segments of the day: daytime (usually defined as 7:00 AM to 11:00 PM) and overnight (usually defined as 11:00 PM to 7:00 AM).

**Quality assessment of the included studies.** The risk of bias of each included study was assessed with the Cochrane Collaboration tool by two independent reviewers(Yuan-yuan WANG and Xiao-lu QIAN) [18]. Each study was scored as having "low risk", "high risk", or "unclear risk" based on the following criteria: randomization, allocation concealment, blinding of participants and personnel, outcome assessment blinding, incomplete data and selective reporting, and other biases. The Jadad score was also calculated (Table 1 and S2 File). We attempted to contact the authors to obtain any missing data. Studies with missing data were excluded when specific outcomes were compared.

## Data synthesis and analysis

The statistical analysis was performed using RevMan 5.3 (RevMan; Copenhagen: The Nordic Cochrane Center, The Cochrane Collaboration, 2014) and STATA (Version 12.0). The effects were expressed as risk ratio (relative risk [RR] or odds ratio [OR]) with 95% confidence interval (CI) and mean difference (MD) with 95% CI for the dichotomous data and continuous data, respectively. A random effects model (M-H heterogeneity) was used to pool data, and $I^2$ statistic >50% and χ2 test ≤ 0.05 were considered to indicate statistical significance. Sensitivity analysis was used to evaluate the robustness of the results. Subgroup analyses and meta-regression analyses were performed to identify the potential sources of heterogeneity. Egger's and Begg's tests were used to evaluate publication bias, but they were not suggested if the number of included studies was less than10 [32]. A p value of less than 0.05 was considered to indicate statistical significance and all reported *p* values were two-tailed.

## Sample size calculation and power analysis

The Type I error probability associated with this test of this null hypothesis is 0.05 and we will use a continuity-corrected chi-squared statistic or Fisher's exact test to evaluate this null hypothesis. This study had only 14.7% power to detect that the HCL group could increase hyperketonemia with sample of 286 patients in the HCL group and 160 patients in the standard of care group. Similarly, the results showed that this study was underpowered (less than 10% power) for identifying the effects of HCL on hypoglycemia, severe hypoglycemia and DKA. The analyses were performed with PS3.0 (Version 3.0, Power and Sample size calculation) software.

## Results

### Study selection and characteristics

Our search strategy initially yielded 2490 articles. After screening the titles and abstracts, 137 articles remained for full-text screening. The total number of participants was increased due to crossover test designs in which participants were included multiple times. Ultimately, a total of 9 articles involving 927 participants were included in this meta-analysis [8, 9, 11, 14, 22, 24, 25, 28, 29] (Fig 1). The average age was 11 years old, and the sample size was greater than 100 in most studies (6/9, 66.7%) [9, 11, 14, 22, 28, 29]. The HCL was only used in the evenings as an overnight closed-loop in only one (1/9, 11.1%) study [25]; in the remaining studies, the HCL system was in use for 24 hours a day (8/9, 88.9%). The average duration of diabetes was six years.

**Risk of bias assessment.** The risks of bias in individual studies and the overall risk of bias are shown in Fig 2 (Fig 2A: risk of bias summary and Fig 2B: risk of bias graph). Although

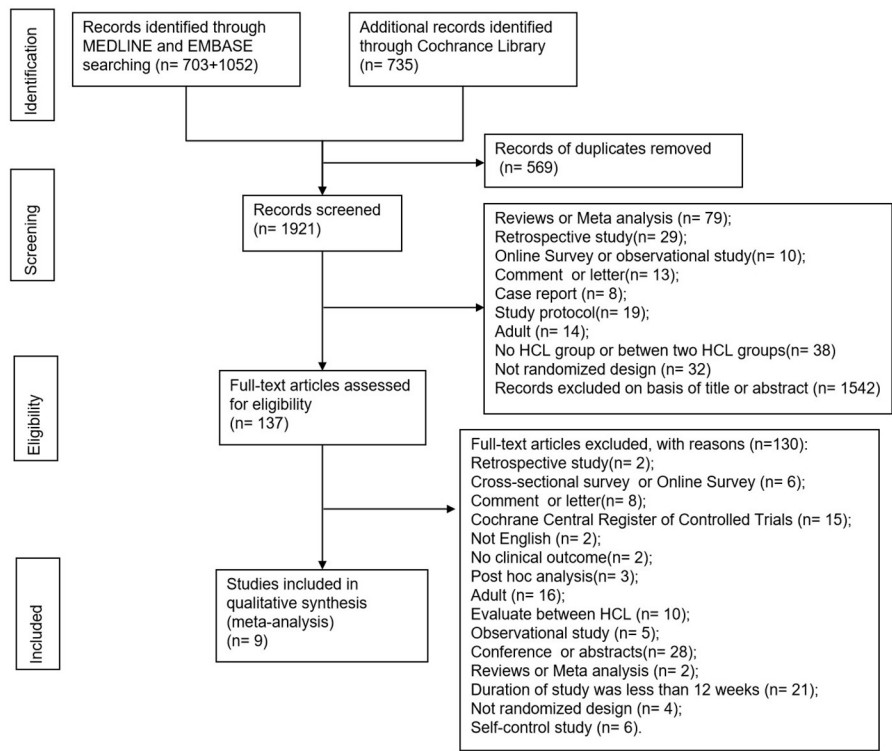

**Fig 1. Flowchart of the study search, selection and inclusion process.**

most studies were randomized-control studies (7/9, 77.8%) [8, 9, 14, 22, 24, 28, 29], the risk of bias was difficult to assess in randomized crossover trials (2/9, 22.2%) [11, 25]. Only one study [24] did not provide specific information on random sequence generation and addressed allocation concealment. The risks of performance bias (8/9, 88.9%) and detection bias (6/9, 66.7%) were considered "unclear" in most studies, as no relevant information was provided.

## Primary outcomes

**HbA1c level.** All included studies [8, 9, 11, 14, 22, 24, 25, 28, 29] with a total of 927 patients compared the efficacy of the HCL group versus the control group to assess the HbA1c level at the end of the study. HCL showed a beneficial effect on HbA1c levels (7.2 ± 1.0 vs. 7.7 ± 1.0%; MD, −0.46, 95% CI, −0.62 to -0.30%; $p$ <0.001, Fig 3) without evidence of heterogeneity between articles ($I^2$ = 40%, $p$ = 0.10; Table 2).

**Sensitivity analysis of HbA1c level.** Sensitivity analyses using the leave one out method revealed that HCLs still showed a beneficial effect on HbA1c levels (S1 Table).

**Publication bias.** Only one studies exceeded the 95% confidence limits in the funnel plot analysis (S1A Fig), however, no significant publication bias was identified based on Begg's ($p$ = 0.917, S1B Fig) and Egger's tests ($p$ = 0.634, S1C Fig).

## Secondary endpoints

**TIR (the percent time in range between 70 and 180 mg/dL).** All included studies [8, 9, 11, 14, 22, 24, 25, 28, 29] with a total of 927 patients compared the safety of the HCL with that of the control group to assess TIR. The percentage of sensor glucose values in TIR was higher

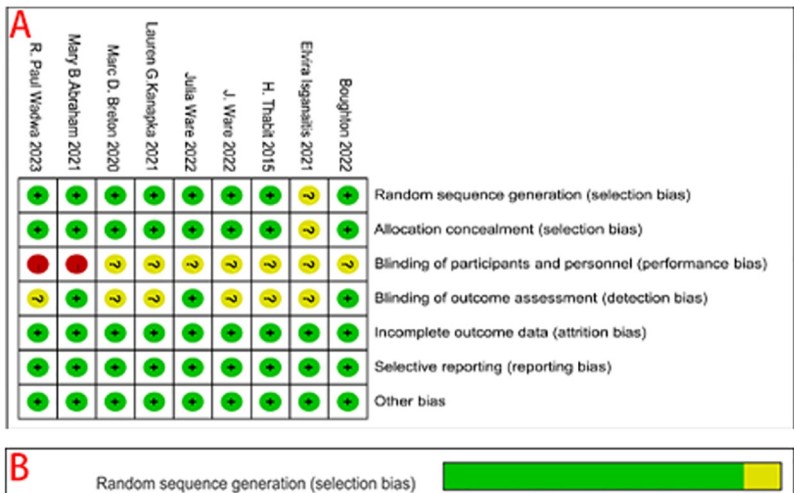

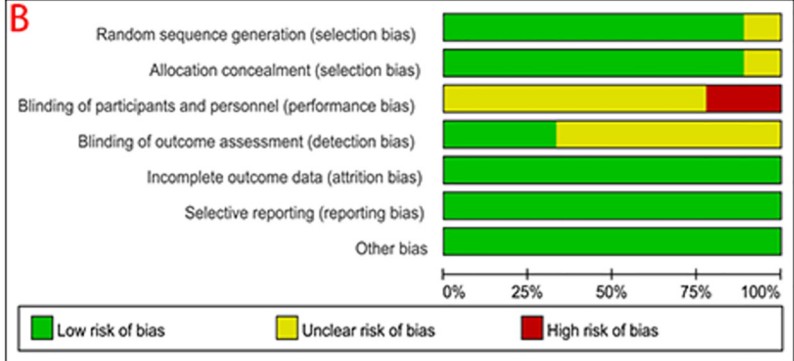

**Fig 2. The risks of bias in individual studies and the overall risk of bias.** A: risk of bias summary and B: risk of bias graph.

in the HCL than in the control overall without evidence of heterogeneity (62.4 ±14.8 vs. 50.6 ±15.2%; MD, 9.96; 95% CI, 7.63 to 12.30%, $p <0.001$; $I^2 = 51\%$, $p = 0.04$; Fig 4A), was higher at night [11, 25, 29] (74.5 ±13.1 vs. 57.8 ±16.6%; MD, 19.16; 95% CI, 13.94 to 24.39%, $p <0.001$; $I^2 = 72\%$, $p = 0.03$; Fig 4B), and during the day [11, 25, 29] (64.7 ± 10.7 vs. 58.0 ±11.3%; MD, 6.39; 95% CI, 2.11 to 10.67%; $p = 0.003$; $I^2 = 60\%$, $p = 0.08$; Fig 4C) (Table 2).

*Sensitivity analysis of TIR.* The level of heterogeneity was significantly decreased when sensitivity analyses were performed that removing the study of "Boughton 2022" [9], "Mary B. Abraham 2021" [22] or "R. Paul Wadwa 2023" [29] and the results were stable (S1 Table).

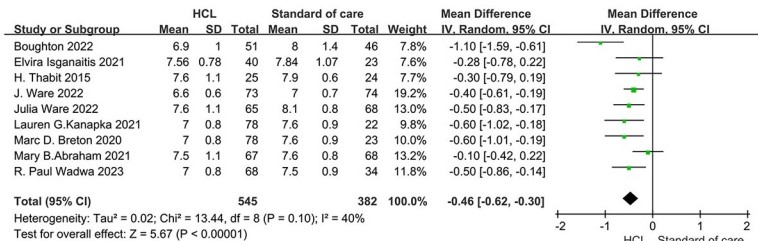

**Fig 3. Forest plot of the risk of HbA1c level in HCL group versus control group.** HCL = Hybrid Closed Loop Systems; RR = relative risk; CI = confidence interval.

**Table 2. Results of glycemic outcomes for the HCL and standard of care groups.**

| Glycemic outcomes | Studies | HCL | | Standard of care | | M-H pooled RR | | Heterogeneity | |
|---|---|---|---|---|---|---|---|---|---|
| | | Mean±SD | Total | Mean±SD | Total | MD (95%CI) | p | $I^2$ (%) | p |
| **HbA1c, %** | [8, 9, 11, 14, 22, 24, 25, 28, 29] | 7.2±1.0 | 545 | 7.7±1.0 | 382 | -0.46(-0.62, -0.30) | < 0.001 | 40 | 0.10 |
| **Overall** | | | | | | | | | |
| Percent of sensor glucose values in range, % | | | | | | | | | |
| < 70 mg/dL | [8, 9, 11, 14, 22, 24, 25, 28, 29] | 3.8±4.5 | 545 | 4.3±4.9 | 382 | -0.22(-0.71, 0.28) | 0.39 | 58 | 0.02 |
| < 54 mg/dL | [9, 11, 14, 22, 24, 25, 28, 29] | 0.73±1.34 | 480 | 0.90±1.40 | 314 | -0.07(-0.19, 0.06) | 0.30 | 64 | 0.007 |
| TIR (70–180 mg/dL) | [8, 9, 11, 14, 22, 24, 25, 28, 29] | 62.4±14.8 | 545 | 50.6±15.2 | 382 | 9.96(7.63, 12.30) | < 0.001 | 51 | 0.04 |
| >180 mg/dL | [8, 9, 11, 14, 22, 24, 25, 28, 29] | 31.9±13.3 | 545 | 48.3±16.1 | 382 | -9.56(-14.17, -4.95) | < 0.001 | 83 | < 0.001 |
| >250 mg/dL | [14, 22, 24, 28, 29] | 9.4±7.9 | 331 | 13.6±9.5 | 170 | -6.82(-11.81, -1.83) | 0.007 | 86 | < 0.001 |
| >300 mg/dL | [9, 11, 22, 24, 28, 29] | 3.2±3.6 | 377 | 5.5±6.8 | 267 | -2.37(-4.03, -0.70) | 0.005 | 76 | 0.001 |
| Glucose level, mg/dL | [8, 9, 11, 14, 22, 24, 25, 28, 29] | 159.9 ±28.2 | 545 | 173.1 ±30.5 | 382 | -14.09(-19.48, -8.70) | < 0.001 | 58 | 0.01 |
| Glucose variability [coefficient of variation %] | [9, 14, 22, 24, 25, 28, 29] | 39.5±6.0 | 407 | 40.3±5.8 | 240 | -0.18(-1.72,1.36) | 0.82 | 71 | 0.002 |
| Daily insulin dose, U/kg per day | [9, 22, 28,29] | 0.90±0.33 | 264 | 0.91±0.33 | 170 | <0.001 (-0.05, 0.06) | 0.94 | < 0.001 | 0.62 |
| **Night** | | | | | | | | | |
| Percent of sensor glucose values in range, % | | | | | | | | | |
| TIR (70–180 mg/dL) | [11, 25, 29] | 74.5±13.1 | 166 | 57.8±16.6 | 132 | 19.16(13.94, 24.39) | < 0.001 | 72 | 0.03 |
| **Day:** | | | | | | | | | |
| TIR (70–180 mg/dL) | [11, 25, 29] | 64.7±10.7 | 166 | 58.0±11.3 | 132 | 6.39(2.11, 10.67) | 0.003 | 60 | 0.08 |

Total = the number of the total patients; RR = relative risk; CI = confidence interval; MD = mean difference; HCL = Hybrid Closed Loop Systems.

Two studies exceeded the 95% confidence limits in the funnel plot analysis and no significant publication bias was identified based on Begg's test ($p = 602$) and Egger's test ($p = 0.133$).

*Subgroup analysis and meta-regression of TIR.* We performed subgroup analyses and meta-regression to identify the potential sources of the heterogeneity. The results showed that age seem to be the source of heterogeneity and the results were stable (S2 and S3 Tables).

**Hypoglycemia.** All included studies [8, 9, 11, 14, 22, 24, 25, 28, 29] patients compared the efficacy of the HCL group versus the control group to assess the percentage of time the sensor glucose was in a hypoglycemic range (defined as <70 mg/dL). HCL did not show a beneficial effect on hypoglycemic time <70 mg/dL (3.8 ± 4.5 vs. 4.3 ± 4.9%; MD, −0.22, 95% CI, −0.71 to 0.28%; $p = 0.39$), and there was evidence of heterogeneity between articles ($I^2 = 58\%$, $p = 0.02$). HCL also did not significantly decrease the overall average percentage of hypoglycemic time <54 mg/dL [9, 11, 14, 22, 24, 25, 28, 29] (0.73 ± 1.34 vs. 0.90 ± 1.40%; MD, −0.07, 95% CI, −0.19 to 0.06%, $p = 0.30$; $I^2 = 64\%$, $p = 0.007$) (Table 2). HCL systems did not seem to be associated with a lower risk of hypoglycemia [22, 25] (2.2% vs. 0%; OR 5.21, 95% CI, 0.24 to 114.41, $p = 0.29$) or severe hypoglycemia [8, 9, 11, 14, 22, 24, 25, 28, 29] (2.6% vs. 1.0%; OR 2.18, 95% CI, 0.77 to 6.23, $p = 0.14$; $I^2 < 0.001\%$, $p = 0.68$), and there was no evidence of heterogeneity (Table 3).

*Sensitivity analysis of hypoglycemia.* Sensitivity analyses using the leave one out method revealed that removing the study of "Boughton 2022" [9] or "Mary B.Abraham 2021" [22]

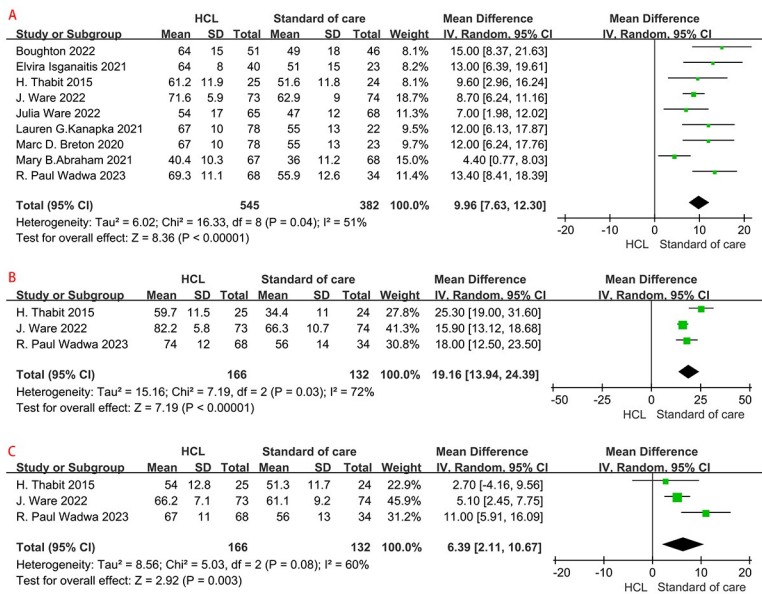

**Fig 4. Forest plot of the risk of TIR in HCL group versus control group.** HCL = Hybrid Closed Loop Systems; RR = relative risk; CI = confidence interval. A: overall, B: at night, C: during the day.

significantly decreased the level of heterogeneity; however, HCL still did not show a beneficial effect on hypoglycemic time <70 and <54 mg/dL (S1 Table).

*Subgroup analysis and meta-regression of time of hypoglycemic time <70 mg/dL.* We performed subgroup analyses and meta-regression to identify the potential sources of the heterogeneity. The results showed that age, type of trial and therapy duration of HCL did not seem to be the sources of heterogeneity (S2 and S3 Tables).

**Hyperglycemia.** The overall percentage of hyperglycemic time was significantly decreased in the HCL group compared to the control group when it was defined as >180 mg/dL [8, 9, 11, 14, 22, 24, 25, 28, 29] (31.9 ± 13.3 vs. 48.3 ± 16.1%; MD, −9.56; 95% CI, −14.17 to −4.95%, $p$ <0.001; $I^2$ = 83%, $p$ <0.001), >250 mg/dL [14, 22, 24, 28, 29] (9.4 ± 7.9 vs. 13.6 ± 9.5%; MD, −6.82; 95% CI, −11.81 to −1.83%; $p$ = 0.007, $I^2$ = 86%, $p$ <0.001) and >300 mg/dL [9, 11, 22, 24, 28, 29] (3.2 ± 3.6 vs. 5.5 ± 6.8%; MD, −2.37; 95% CI, −4.03 to -0.70%, $p$ = 0.005; $I^2$ = 76%, $p$ = 0.001), and there was significant heterogeneity (Table 2).

HCL did not seem to be associated with a lower risk of hyperketosis [8, 14, 24, 25, 28, 29] (13.0% vs. 11.3%; OR 1.19, 95% CI, 0.63 to 2.25, $p$ = 0.60), and DKA [8, 9, 14, 22, 24, 25, 28, 29] (0.13% vs. 0%; OR 2.18, 95% CI, 0.43 to 11.00, $p$ = 0.34), and there was no significant heterogeneity (Table 3).

**Table 3. Results of outcomes for the HCL and standard of care groups.**

| Glycemic outcomes | Studies | HCL | | Standard of care | | Overall event rates (%) | M-H pooled OR | | Heterogeneity | |
|---|---|---|---|---|---|---|---|---|---|---|
| | | event+ | Total | event+ | Total | | OR (95%CI) | $p$ | $I^2$ (%) | $p$ |
| Hypoglycemia | [22, 25] | 2(2.2%) | 92 | 0(0%) | 92 | 1.1% | 5.21(0.24, 114.41) | 0.29 | NA | NA |
| Severe hypoglycemia | [8, 9, 11, 14, 22, 24, 25, 28, 29] | 14(2.6%) | 545 | 4(1.0%) | 382 | 1.9% | 2.18(0.77, 6.23) | 0.14 | <0.001 | 0.68 |
| Diabetic ketoacidosis (DKA) | [8, 9, 14, 22, 24, 25, 28, 29] | 6(0.13%) | 472 | 0(0%) | 240 | 0.84% | 2.18(0.43, 11.00) | 0.34 | <0.001 | 0.99 |
| Hyperketonemia | [8, 14, 24, 25, 28, 29] | 46(13.0%) | 354 | 22(11.3%) | 194 | 12.4% | 1.19(0.63, 2.25) | 0.60 | 7 | 0.37 |

HCL, Hybrid Closed Loop Syst

*Sensitivity analysis of hyperglycemia.* Sensitivity analyses using the leave one out method revealed that the results were stable with respect to hyperglycemic time >180 mg/dL, >250 and >300 mg/dL (S1 Table).

*Subgroup analysis and meta-regression of time of hyperglycemic time >180 mg/dL.* We performed subgroup analyses and meta-regression to identify the potential sources of the heterogeneity. The results showed that age, type of trial and therapy duration of HCL did not seem to be the sources of heterogeneity (S2 and S3 Tables).

**Glucose control and insulin delivery.** The mean glucose level[8, 9, 11, 14, 22, 24, 25, 28, 29] (159.9 ± 28.2 vs. 173.1 ± 30.5 mg/dL; MD, −14.09; 95% CI, −19.48 to -8.70 mg/dL, *p* <0.001; $I^2$ = 58%, *p* = 0.01) was significantly decreased by HCL, however, no significant difference was found in the coefficient of variation of sensor glucose[9, 14, 22, 24, 25, 28, 29] (39.5 ± 6.0 vs. 40.3 ± 5.8%; MD, −0.18; 95% CI, −1.72 to 1.36%, *p* = 0.82; $I^2$ = 71%, *p* = 0.002) and daily insulin dose [9, 22, 28, 29] (0.90 ± 0.33 vs. 0.91 ± 0.33 U/kg per day; MD, <0.001; 95% CI, −0.05 to 0.06 U/kg per day; *p* = 0.94; $I^2$ <0.001, *p* = 0.62) between the HCL group and the control group (Table 2).

*Sensitivity analysis of the mean glucose level.* The level of heterogeneity was significantly decreased when sensitivity analyses were performed that removing the study of "Boughton 2022" [9] or "Mary B.Abraham 2021" [22] and the results were stable (S1 Table).

## Discussion

The pooled results of this meta-analysis showed that HCL exert a beneficial effect on HbA1c management without significant heterogeneity as compared to the standard of care in adolescents and children with T1D when therapy duration of HCL was not less than three months, we should notice that significant heterogeneity was observed in previous meta analyses [33–36]. HCL increased the percentage of time in TIR and decreased hyperglycemic time, which was similar to a recent meta-analysis [26, 27] that better TIR, HbA1c, and less hypoglycemia in HCL in patients with T1D were found. However, the recent important studies [8, 9, 11, 22, 24, 28, 29] were not included in those recent meta-analysis [26, 27], adults were included and the study period was different from 48 h to 6 months.

The management of T1D is further complicated in young individuals, who have unique physiological, behavioral and developmental factors. Young children have higher insulin sensitivity, higher variability in insulin requirements, and more unpredictable eating and activity patterns than adults [36]. HCL have the advantage of delivering insulin in a glucose-responsive manner [37], which is expected to benefit clinical outcomes. Long-term glycemic variability has been reported to be involved in increasing cardiovascular risk, retinopathy and renal failure [38]. In this meta-analysis, HCL showed a beneficial effect on HbA1c management, with a 0.5% decrease compared to the compared to standard of care in adolescents and children with T1D, which was more than the recent study of Boughton with 0.3% decreasing of HbA1c by HCL systems [9]. In a recent pivotal study assessing the HCL over 3 months, the mean HbA1c was improved by 0.7% in children and 0.4% in adults [18]. In a recent 6-month, multicenter, randomized study, HbA1c levels were also significantly decreased and TIR was significantly increased in the HCL group compared to the control group among patients with T1D aged 14–61 years [13]. Compared with the control group, there was a 0.33% reduction in HbA1c levels among the individuals aged 14 to 71 years old using the HCL [13]. Compared to baseline, HbA1c levels improved from 7.3% to 6.8% in the adult group and 7.7% to 7.1% in the adolescent group after 3 months of HCL administration [20].

We only included studies with a duration of more than 3 months, as the HbA1c levels reflect the average blood glucose in 3 months. Previous studies have also shown that glycemic

variability and HbA1c levels are reduced in adolescents and adults [20,21] after at least 3 months of HCL administration. Another study even showed that glucose variability was significantly decreased throughout the 24-h day by HCL when compared with baseline [39].

The results of the current meta-analysis were also in agreement with previous meta-analyses [26, 33–35] and many studies [20, 38, 40] that reported that the TIR was significantly increased and the percentage of time in the hyperglycemic range was significantly lower for HCL use than for the standard of care in children and adolescents. The TIR was reported to be increased by approximately 5.7 [40]-12% [34] in the HCL group compared to the standard of care. Our meta-analysis also showed that HCL led to a 11.1% increase in the TIR compared with the standard of care in adolescents and children with T1D over longer periods. A recent meta-analysis showed an 11.73% improvement in TIR based on 25 studies with a total of 504 patients [36]. Another meta-analysis also showed an 11.06% [33] and 9.62% [41] improvement in TIR when comparing HCL to non-closed-loop systems. Not only is TIR generally improved, but TIR during exercise was also improved by 6.18% after HCL administration, especially in children and adolescents [36].

HCL systems did not exert a beneficial effect on hypoglycemic time compared to the standard of care in adolescents and children with T1D, which not consistent with the recent meta-analysis [26]. We should note that the current HCL requires a minimum total daily insulin dose for optimal system performance; however, the total daily dose is usually very low in adolescents and children. This may limit the benefits of closed-loop treatment due to the high variability of such small dose absorption [42]. The variability in blood glucose was higher during the nighttime in young individuals [37], and nocturnal hypoglycemia was expected to be reduced by HCL due to the automation of insulin delivery in response to real-time sensor glucose levels. The beneficial effect of HCL on the nocturnal hypoglycemic time (defined as <70 mg/dL and <54 mg/dL) was also not noted in this meta-analysis. This is also likely related to the ability of HCL to immediately r reduce postprandial hyperglycemia, which is limited by the pharmacodynamics and pharmacokinetics of rapid-acting insulins [43] and even increases the risk of delayed hypoglycemia as a result of increasing algorithm-driven insulin delivery. The rates of hypoglycemia and severe hypoglycemia were not increased by HCL in this meta-analysis. Second, the sensor accuracy of HCL was decreased due to the physiological lag time between blood glucose and interstitial glucose during daytime physical activity, irrespective of the quality of the algorithm [44]. The basal insulin dose has also been found to either decrease [20] or increase [45] during HCL use when compared with controls in some studies. This meta-analysis showed no significant difference in daily insulin dose when HCL was used.

Hyperglycemia was marked in children with type 1 diabetes. One study reported that sensor glucose readings >180 mg/dL even occurred in nearly half of the overnight hours among children with T1D [40]. Parents are always afraid of hypoglycemia and thus might allow hyperglycemia to alleviate worries of dangerous hypoglycemia, especially at night [46]. HCL may reduce parental fears of hypoglycemia at night, which may prevent hyperglycemia-permissive behaviors. Our results showed that HCL led to a greater decrease in time in the hyperglycemia range when it was defined as >180 mg/dL, >250 mg/dL and >300 mg/dL; however, significant heterogeneity remained. However, we found that HCL did not seemed to be associated with a lower risk of hyperketonemia and DKA.

## Strengths and limitations

**Strengths of this meta-analysis.** For the first time, we included the largest sample (n = 825) and the most rigorous studies as most study designs were randomized trials. HCL studies in adolescents and children have always examined small samples and have had a

therapy durations [47]. Second, we performed a comprehensive study to assess the effect of long-term use of HCL on glycemic outcomes in children and adolescents.

**Limitations of this meta-analysis.** First, the therapy duration of HCL (between 12 weeks and 24 months) and mean age of included patient (between 2 and 15 years old) varied. In addition, the number of included studies was relatively small, thus, the results might be biased.

## Conclusions

HCL had a beneficial effect on HbA1c management and TIR without increased hypoglycemic time as compared to standard of care in adolescents and children with T1D when therapy duration of HCL was not less than three months.

## Supporting information

**S1 File. Search strategies for this study.**
(DOC)

**S2 File. The Jadad scale for assessing the methodological quality of clinical trials.**
(DOC)

**S1 Fig. Publication bias analysis of HbA1c level.** A: A funnel plot of the risk of HbA1c level. B: Begg's test; C: Egger's test.
(TIF)

**S1 Table. Sensitivity analysis for the outcome of HbA1c level, glucose level and percent of sensor glucose values.** HbA1c, Glycated hemoglobin level; MD, mean difference; CI, Confidence interval.
(DOC)

**S2 Table. Subgroup analysis for the outcome of glucose level and percent of sensor glucose values.** HCL, Hybrid Closed Loop Systems.
(DOC)

**S3 Table. Meta-regression for the outcome of glucose level and percent of sensor glucose values.** HCL, Hybrid Closed Loop Systems.
(DOC)

**S1 Checklist. PRISMA guidelines checklist.**
(DOC)

## Author Contributions

**Data curation:** Hui-min YING, Fang TIAN, Xiao-lu QIAN.

**Formal analysis:** Fang TIAN.

**Investigation:** Hui-min YING, Xiao-lu QIAN.

**Writing – original draft:** Yuan-yuan WANG, Zhen-feng Zhou.

**Writing – review & editing:** Yuan-yuan WANG, Zhen-feng Zhou.

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
