## [Decision Letter · Decision Letter 0]

26 Jun 2024

PONE-D-24-12735Three months use of Hybrid Closed Loop Systems Could Improve Glycated Hemoglobin Levels in Adolescents and Children with Type 1 Diabetes: A Meta-AnalysisPLOS ONE

Dear Dr. ZHOU,

Thank you for submitting your manuscript to PLOS ONE. After careful consideration, we feel that it has merit but does not fully meet PLOS ONE’s publication criteria as it currently stands. Therefore, we invite you to submit a revised version of the manuscript that addresses the points raised during the review process.

We look forward to receiving your revised manuscript.

Kind regards,

Timotius Ivan Hariyanto, M.D.

Academic Editor

PLOS ONE

Additional Editor Comments (if provided):

Reviewers' comments:

Reviewer's Responses to Questions

**Comments to the Author**

1. Is the manuscript technically sound, and do the data support the conclusions?

Reviewer #1: Yes

Reviewer #2: Yes

2. Has the statistical analysis been performed appropriately and rigorously? 

Reviewer #1: Yes

Reviewer #2: Yes

3. Have the authors made all data underlying the findings in their manuscript fully available?

Reviewer #1: Yes

Reviewer #2: Yes

4. Is the manuscript presented in an intelligible fashion and written in standard English?

Reviewer #1: Yes

Reviewer #2: No

5. Review Comments to the Author

Reviewer #1: The article is good meta-analysis of Closed loop insulin pump use and benefits in the management of Type 1 Diabetes.Extensive references are an asset of any review article. The longterm benefits like A1c reductions and hypoglycaemias are important aspects of a pump use in children & adolescent with Type 1 Diabetes.The total number of subjects in the meta-analysis is good enough to get a tangible and acceptable concusions.

Reviewer #2: The authors conducted a meta-analysis assessing variations in glycated hemoglobin due to the use of automated insulin delivery systems. The study is methodologically sound, and the results confirm the benefits of these innovative technological devices for diabetes management. However, there is a significant concern regarding the authors' decision to include only randomized clinical trials (RCTs). In the field of diabetes technology, some of the most impactful findings often stem from real-world studies. Can the authors provide a rationale for this choice?

Additionally, there are several typos throughout the text. A thorough revision of the style and language is recommended to enhance the overall quality of the manuscript.

6. PLOS authors have the option to publish the peer review history of their article (what does this mean?). If published, this will include your full peer review and any attached files.

Reviewer #1: No

Reviewer #2: No

---

## [Author Response · Author response to Decision Letter 0]

27 Jun 2024

Dear Editor in Chief:

Thanks for your kind suggestions. I have sincerely considered your and the reviewers’ comments.

Thanks again for your time and effort expending on this paper.

Best wishes.

Yours sincerely, 

Zhen-feng Zhou

1.1. Please ensure that your manuscript meets PLOS ONE's style requirements, including those for file naming. The PLOS ONE style templates can be found at 

Answer: we have revised the manuscript according to the PLOS ONE's style requirements.

1.2. Please provide a complete Data Availability Statement in the submission form, ensuring you include all necessary access information or a reason for why you are unable to make your data freely accessible. If your research concerns only data provided within your submission, please write "All data are in the manuscript and/or supporting information files" as your Data Availability Statement. 

Answer: we have revised the Data Availability Statement as: “All data are in the manuscript and supporting information files.” 

1.3. Please review your reference list to ensure that it is complete and correct. If you have cited papers that have been retracted, please include the rationale for doing so in the manuscript text, or remove these references and replace them with relevant current references. Any changes to the reference list should be mentioned in the rebuttal letter that accompanies your revised manuscript. If you need to cite a retracted article, indicate the article’s retracted status in the References list and also include a citation and full reference for the retraction notice.

Answer: we have reviewed and revised our reference list.

2. Review Comments to the Author

2.1Reviewer #1: The article is good meta-analysis of Closed loop insulin pump use and benefits in the management of Type 1 Diabetes.Extensive references are an asset of any review article. The longterm benefits like A1c reductions and hypoglycaemias are important aspects of a pump use in children & adolescent with Type 1 Diabetes.The total number of subjects in the meta-analysis is good enough to get a tangible and acceptable concusions.

Answer: Thank you so much!

2.2 Reviewer #2: The authors conducted a meta-analysis assessing variations in glycated hemoglobin due to the use of automated insulin delivery systems. The study is methodologically sound, and the results confirm the benefits of these innovative technological devices for diabetes management. However, there is a significant concern regarding the authors' decision to include only randomized clinical trials (RCTs). In the field of diabetes technology, some of the most impactful findings often stem from real-world studies. Can the authors provide a rationale for this choice?

Additionally, there are several typos throughout the text. A thorough revision of the style and language is recommended to enhance the overall quality of the manuscript.

Answer: the purpose of this Meta-Analysis is investigate that 3 months use of HCL could improve glycated hemoglobin levels in adolescents and children with T1D when compared to standard of care. As we know, the best method to compare the advantages and disadvantages of new therapies with standard therapies is a randomized controlled trial, so we only include randomized clinical trials (RCTs). The phenomenon that makes it difficult to interpret the therapeutic effect is called the "confounding effect", and randomization can prevent this effect from occurring. Randomization can ensure that patients participating in different treatment trials have roughly the same physiological characteristics, so there is comparability of data between different groups. If there is a difference in the treatment effect between the two groups at this time, it can be considered that it is caused by the difference in the treatment itself. Therefore, randomization is a crucial step in directly comparing the therapeutic effects of different therapies. 

The manuscript was revised by one of the companies that Elsevier recommends for English translation services before we submit it.

---

## [Decision Letter · Decision Letter 1]

17 Jul 2024

PONE-D-24-12735R1Three months use of Hybrid Closed Loop Systems Could Improve Glycated Hemoglobin Levels in Adolescents and Children with Type 1 Diabetes: A Meta-AnalysisPLOS ONE

Dear Dr. ZHOU,

Thank you for submitting your manuscript to PLOS ONE. After careful consideration, we feel that it has merit but does not fully meet PLOS ONE’s publication criteria as it currently stands. Therefore, we invite you to submit a revised version of the manuscript that addresses the points raised during the review process.

Please remove the word "could" from the title as the reviewer has suggested before I can formally accept this manuscript.

We look forward to receiving your revised manuscript.

Kind regards,

Timotius Ivan Hariyanto, M.D.

Academic Editor

PLOS ONE

Journal Requirements:

Additional Editor Comments:

Please remove the word "could" from the title as the reviewer has suggested before I can formally accept this manuscript.

Reviewers' comments:

Reviewer's Responses to Questions

**Comments to the Author**

1. If the authors have adequately addressed your comments raised in a previous round of review and you feel that this manuscript is now acceptable for publication, you may indicate that here to bypass the “Comments to the Author” section, enter your conflict of interest statement in the “Confidential to Editor” section, and submit your "Accept" recommendation.

Reviewer #1: All comments have been addressed

Reviewer #2: (No Response)

2. Is the manuscript technically sound, and do the data support the conclusions?

Reviewer #1: Yes

Reviewer #2: (No Response)

3. Has the statistical analysis been performed appropriately and rigorously? 

Reviewer #1: Yes

Reviewer #2: (No Response)

4. Have the authors made all data underlying the findings in their manuscript fully available?

Reviewer #1: Yes

Reviewer #2: (No Response)

5. Is the manuscript presented in an intelligible fashion and written in standard English?

Reviewer #1: Yes

Reviewer #2: (No Response)

6. Review Comments to the Author

Reviewer #1: All the quiries of reviewers has been clarified & corrected satisfactorily.The article may be accepted for publication in the latest version.

Reviewer #2: I suggest changing the title. Particularly, removing the word "could" makes the title more impactful.

7. PLOS authors have the option to publish the peer review history of their article (what does this mean?). If published, this will include your full peer review and any attached files.

Reviewer #1: No

Reviewer #2: No

---

## [Author Response · Author response to Decision Letter 1]

18 Jul 2024

Dear Editor in Chief:

I have sincerely considered your and the reviewers’ comments.

We hope that our revised version will be satisfactory for publication in PLOS ONE. Great thanks to you and the referee for the time and effort you expend on this paper.

Best wishes.

Yours sincerely, 

Zhen-feng Zhou

Answer: we have reviewed and revised our reference list. We remove the reference 43 that have been retracted and delete the discussion of reference 43. Page 12 line 20-23 in Revised Manuscript with Track Changes.

And we replace references 3 with relevant current references.

2.Additional Editor Comments:

Please remove the word "could" from the title as the reviewer has suggested before I can formally accept this manuscript.

Answer: we have removed the word "could" from the title as the reviewer has suggested. Page 1 line 1.

Review Comments to the Author

Dear reviewer, thank you very much for your valuable recommendations. Great thanks to you and the referee for the time and effort you expend on this paper.

Thank you so much, you are so kind!

Reviewer #1: All the quiries of reviewers has been clarified & corrected satisfactorily.The article may be accepted for publication in the latest version.

Answer: Thank you so much, you are so kind!

Reviewer #2: I suggest changing the title. Particularly, removing the word "could" makes the title more impactful.

Answer: thank you very much for your valuable recommendations. We have removed the word "could" from the title as you suggested. Page 1 line 1.

---

## [Editor Report · Decision Letter 2]

19 Jul 2024

Three months use of Hybrid Closed Loop Systems Improves Glycated Hemoglobin Levels in Adolescents and Children with Type 1 Diabetes: A Meta-Analysis

PONE-D-24-12735R2

Dear Dr. ZHOU,

We’re pleased to inform you that your manuscript has been judged scientifically suitable for publication and will be formally accepted for publication once it meets all outstanding technical requirements.

Kind regards,

Timotius Ivan Hariyanto, M.D.

Academic Editor

PLOS ONE
---

## [Editor Report · Acceptance letter]

2 Aug 2024

PONE-D-24-12735R2 

PLOS ONE

Dear Dr. ZHOU, 

I'm pleased to inform you that your manuscript has been deemed suitable for publication in PLOS ONE. Congratulations! Your manuscript is now being handed over to our production team.

Kind regards, 

on behalf of

Dr. Timotius Ivan Hariyanto 

Academic Editor

PLOS ONE